# Optimizing nnU-Net with OpenVINO for Fast CPU Inference in Abdominal Organ Segmentation

Yannick Kirchhoff[1,2,3*], Ashis Ravindran[1*], Maximilian Rokuss[1,3],
Benjamin Hamm[1,4], Constantin Ulrich[1,4,5], Klaus Maier-Hein[1,6†], and
Fabian Isensee[1,7†]

[1] German Cancer Research Center (DKFZ) Heidelberg, Division of Medical Image Computing, Heidelberg, Germany
[2] HIDSS4Health - Helmholtz Information and Data Science School for Health, Karlsruhe/Heidelberg, Germany
[3] Faculty of Mathematics and Computer Science, Heidelberg University, Heidelberg, Germany
[4] Medical Faculty Heidelberg, Heidelberg University, Heidelberg, Germany
[5] National Center for Tumor Diseases (NCT), NCT Heidelberg, A partnership between DKFZ and University Medical Center Heidelberg
[6] Pattern Analysis and Learning Group, Department of Radiation Oncology, Heidelberg University Hospital, Heidelberg, Germany
[7] Helmholtz Imaging, DKFZ, Heidelberg, Germany
{yannick.kirchhoff, ashis.ravindran}@dkfz-heidelberg.de

**Abstract.** Accurate segmentation of abdominal organs in pathological computed tomography (CT) scans is crucial for diagnosis and treatment planning. However, this task is challenging due to the diversity of organ appearances and sizes, as well as the computational limitations in clinical settings. Task 2 of the FLARE 2024 challenge was launched to encourage the development of algorithms capable of efficient abdominal organ segmentation under strict resource constraints, specifically focusing on CPU-based inference without access to GPUs. In this paper, we describe our contribution to this challenge by utilizing nnU-Net with optimizations for efficient CPU-based inference using OpenVINO. We resampled the CT scans to an isotropic low resolution to balance segmentation accuracy and computational efficiency. Our method achieved an average Dice Similarity Coefficient (DSC) of 76.8% and an average Normalized Surface Dice (NSD) of 80.5% on the public validation set, with an average running time of 26 seconds per case. These results demonstrate that our approach effectively addresses the challenges of efficient organ segmentation under resource constraints, and underscore the potential for deploying such methods in real-world clinical environments where computational resources are limited.

**Keywords:** FLARE Challenge · Organ Segmentation · nnU-Net · OpenVINO.

---

[*] Equal contribution
[†] Shared last authorship

## 1   Introduction

Accurate organ and lesion segmentation in medical imaging is crucial for improving diagnostic accuracy, treatment planning, and monitoring the progression of diseases. In recent years, segmentation challenges in medical imaging have driven significant advancements in algorithm development, particularly in the field of abdominal cancer segmentation. However, the task of abdominal organ segmentation on pathological scans presents unique challenges due to the wide variety of cancer types, lesion sizes, and corresponding differences in appearances of organs.

Task 2 of the FLARE 2024 challenge builds on earlier iterations of the FLARE challenge, focusing on abdominal organ segmentation on pathological computed tomography (CT) scans. The provided dataset spans 2,050 CT scans: 50 with full annotations for 13 organ classes, and the remaining 2,000 scans with pseudo-labels created by the winning solution from the 2023 version of the FLARE challenge. The difficulty in this task mainly lies in handling the pseudo-labeled data split and the limitations imposed by CPU-based inference with strict time and resource constraints.

A popular strategy for handling unlabeled data is pseudo-label generation, which creates inferred labels for unlabeled data points based on the model's current predictions. This technique has been widely adopted in semi-supervised learning tasks, particularly for large unlabeled datasets. Notably, it was successfully implemented by the winners of the 2022 FLARE challenge [9,16], who employed pseudo-labeling to maximize performance on unlabeled datasets. Applying this strategy – either by actively generating pseudo-labels or using the pseudo-labels generated by the winning solution from FLARE 2023 – is therefore straightforward.

This manuscript describes our approach for abdominal organ segmentation in the presence of lesions in Task 2 of the FLARE 2024 challenge. We employ nnU-Net [10] with modifications for efficient inference on CPUs using OpenVINO, an open-source toolkit for optimizing and deploying deep learning models, to adhere to resource and time constraints during inference. We only focus on optimizing the nnU-Net model for fast inference and performance using the fully labeled subset of the provided data, and do not use the unlabeled / pseudolabeled data at all.

## 2   Method

Our contribution builds upon the state-of-the-art nnU-Net framework [10]. Due to the time and resource constraints imposed during inference, we cannot use the proposed default U-Net configuration, nor the newly proposed ResEncL configuration [11]. We employ just-in-time (JIT) compiled functions to accelerate prediction speed, ensuring our model remains efficient even on resource-constrained devices like laptops and edge devices.

### 2.1   Proposed Method

**Preprocessing**  All images are normalized according to nnU-Net's CT-Normalization, i.e. intensity clipping to $[-958, 270]$ followed by subtracting $-97.3$ and dividing by 137.8. In the default configuration, all images would be resampled to the median spacing $[2.5mm, 0.8mm, 0.8mm]$, however CPU-based inference with this high resolution is not practical within the time limit. We therefore experiment with two spacing configurations, an isotropic spacing of $[2.5mm, 2.5mm, 2.5mm]$ and, as an ablation, a more typical low-resolution spacing of $[5mm, 1.6mm, 1.6mm]$.

**Training:** nnU-Net generates a default configuration with a patch size of 40x224x192, batch size of 2 and a U-Net with 6 resolution stages. We use the same patch size for our low-resolution trainings in order to avoid patch sizes considerably larger than the median image size of 97x512x512 at full resolutions. We increase the batch size to 4. Figure 1 shows a schematic overview of the generated network architeture.

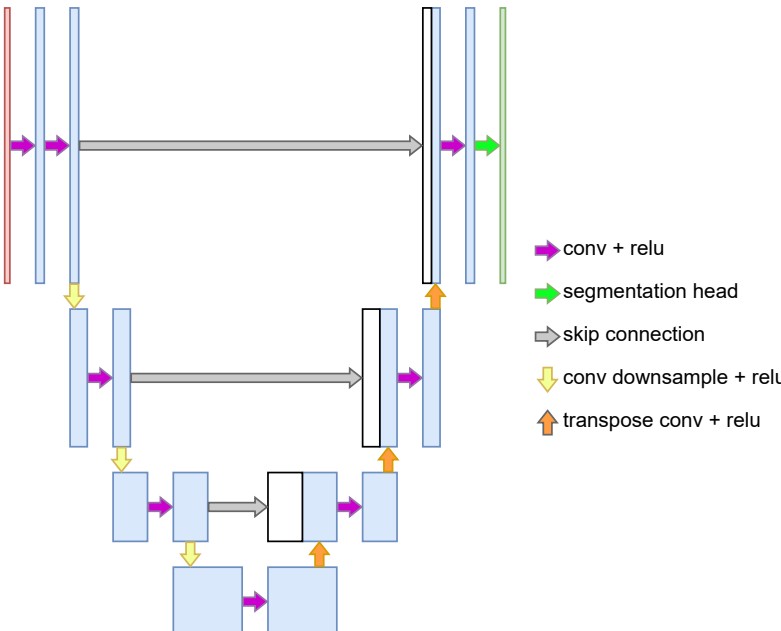

**Fig. 1.** Schematic network architecture of the U-Net created by nnU-Net's default configuration.

**Inference:** nnU-Net's inference pipeline is not optimized for CPU-based inference as required in this challenge. We therefore JIT compile our model using OpenVINO[3] minimizing latency and maximizing the throughput, ensuring that our approach is viable for CPU-based scenarios. OpenVINO is an open-source toolkit for optimizing and deploying deep learning models from cloud to edge. This optimization toolchain converts our trained model into an optimized intermediate representation, enabling efficient execution on Intel CPUs and other compatible hardware. By leveraging OpenVINO, we achieve significant improvements in inference speed by a factor of $\sim$20, making our solution practical for use in real-world clinical settings where computational resources are limited.

In addition to this major change, we disable all test time augmentations, and calculate the argmax directly on the raw logits instead of the softmax probabilities. Furthermore, we swap the default *skimage*-based resampling function for the much faster *torch* resampling, significantly speeding up segmentation export in exchange for a slight loss in performance.

## 3    Experiments

### 3.1    Dataset and evaluation measures

The dataset is curated from more than 40 medical centers under license permission, including TCIA [2], LiTS [1], MSD [19], KiTS [6,8,7], autoPET [5,4], AMOS [12], AbdomenCT-1K [18], TotalSegmentator [20], and past FLARE challenges [15,16,17]. The training set includes 2050 abdomen CT scans where 50 CT scans have complete labels and 2000 CT scans have no labels. The validation and testing sets include 250 and 300 CT scans, respectively. The annotation process used ITK-SNAP [22], nnU-Net [10], MedSAM [13], and Slicer Plugins [3,14].

The evaluation metrics encompass two accuracy measures—Dice Similarity Coefficient (DSC) and Normalized Surface Dice (NSD)—alongside one efficiency measures—runtime. These metrics collectively contribute to the ranking computation. During inference, a GPU is not available and the algorithm should only rely on a CPU.

### 3.2    Implementation details

**Environment settings** The development environments and requirements are presented in Table 1.

**Training protocols** We trained our models only on the fully labeled subset of the provided dataset. We used the default nnU-Net pipeline of data augmentations, consisting of spatial transformations — i.e., rotations, mirroring — and intensity transformations, without further modifications. The final models were selected by expected inference times and performance on the public validation set.

---

[3] https://github.com/openvinotoolkit/openvino  and  https://docs.openvino.ai/

**Table 1.** Development environments and requirements.

| | |
|---|---|
| System | Ubuntu 20.04 |
| CPU | AMD Ryzen 9 3900X processor |
| RAM | 64GB DDR4-3600 RAM |
| GPU (number and type) | One NVIDIA RTX3090 GPU with 24GB VRAM |
| CUDA version | 12.1 |
| Programming language | Python 3.11 |
| Deep learning framework | torch 2.4.0 |

**Table 2.** Training protocols.

| | |
|---|---|
| Network initialization | random |
| Batch size | 4 |
| Patch size | $40 \times 224 \times 192$ |
| Total epochs | 1000 |
| Optimizer | SGD |
| Initial learning rate (lr) | 1e-2 |
| Lr decay schedule | PolyLR Scheduler |
| Loss function | Soft Dice loss + Cross Entropy loss |
| Number of model parameters | 30.71M |

### 3.3 Test Set Submission

Task 2 of the FLARE challenge allowed for only one submission to the final test set. We therefore submitted the model trained with isotropic spacing of 2.5mm, which showed better performance than the half resolution model on the public validation set (see Table 3).

## 4 Results and Discussion

### 4.1 Quantitative results on validation set

The results of the final submission on the public validation set are shown in Table 3. The model trained with isotropic low-resolution spacing considerably outperforms the model trained with half resolution. This is likely due to the large spacing of 5mm in the half resolution model, which leads to a loss of 3D context compared to the 2.5mm spacing of the isotropic low-resolution model.

### 4.2 Qualitative results on validation set

Figure 2 shows qualitative results of the submitted method and the ablation on four cases from the public validation set. The submitted method generally performs well on most abdominal organs. The ablation with half resolution performs significantly worse, even on comparably large organs like the spleen. The last two

**Table 3.** Quantitative evaluation results of the submitted method and an ablation on the public validation set.

| Target | Public Validation | | Public Validation (Ablation) | |
|---|---|---|---|---|
| | DSC (%) | NSD (%) | DSC (%) | NSD (%) |
| Liver | **95.1 ± 8.7** | **92.1 ± 13.1** | 90.0 ± 8.8 | 79.9 ± 13.6 |
| Right Kidney | **79.3 ± 29.4** | **77.6 ± 29.1** | 61.4 ± 32.8 | 57.7 ± 30.4 |
| Spleen | **85.7 ± 25.2** | **83.1 ± 27.0** | 54.4 ± 36.0 | 54.6 ± 34.1 |
| Pancreas | **77.6 ± 9.6** | **84.2 ± 9.5** | 36.7 ± 18.4 | 44.8 ± 20.2 |
| Aorta | **92.6 ± 7.2** | **94.2 ± 9.8** | 75.8 ± 19.6 | 72.3 ± 20.5 |
| Inferior vena cava | **81.8 ± 18.6** | **81.0 ± 20.4** | 56.0 ± 22.5 | 48.8 ± 21.5 |
| Right adrenal gland | **67.7 ± 22.1** | **82.6 ± 26.0** | 38.3 ± 29.5 | 47.5 ± 34.7 |
| Left adrenal gland | **65.7 ± 27.9** | **78.1 ± 31.1** | 16.9 ± 25.4 | 22.3 ± 30.6 |
| Gallbladder | **56.1 ± 39.8** | **55.9 ± 40.7** | 35.0 ± 36.5 | 32.5 ± 34.8 |
| Esophagus | **77.3 ± 15.5** | **87.4 ± 15.1** | 52.8 ± 25.8 | 61.3 ± 27.3 |
| Stomach | **80.0 ± 20.2** | **80.6 ± 18.4** | 58.0 ± 28.6 | 55.1 ± 24.9 |
| Duodenum | **64.5 ± 16.8** | **78.9 ± 14.9** | 29.4 ± 19.5 | 46.0 ± 20.9 |
| Left kidney | **75.0 ± 33.9** | **70.8 ± 33.6** | 41.5 ± 36.1 | 42.4 ± 33.9 |
| Average | **76.8 ± 11.7** | **80.5 ± 12.7** | 49.7 ± 15.3 | 51.2 ± 16.0 |

rows show failure modes of the submitted model. The confusion of left and right kidneys is a sign of missing context to learn left-right symmetries, which is likely due to a combination of patch size and low resolution. The prediction of organs in the legs hints at insufficient data during training and could likely be improved by including the unlabeled / pseudo-labeled data for training.

### 4.3   Segmentation efficiency results on validation set

Table 4 shows the running time of the submitted method on 8 selected cases from the public validation set. The model complies with the time limit for all of the 8 cases. However, as the testing was performed on a significantly better CPU, it is expected that the model might exceed the time limit for exceptionally large cases in the final testing. Our approach prioritizes efficiency, making it suitable for deployment on edge devices and in resource-constrained environments. By utilizing just-in-time (JIT) compiling, we achieve significant reductions in computational load and inference time, as demonstrated in Table 4.

### 4.4   Results on final testing set

Tables 5 and 6 show the final results for segmentation performance and efficiency on the three test set cohorts, respectively. The model shows a large drop in performance for the asian and north american cohorts with respect to the european cohort, indicating a lack of generalization performance.

### 4.5   Limitation and future work

The main limitation of our work is that we did not use the unlabeled data for model development and training, which would likely improve performance and

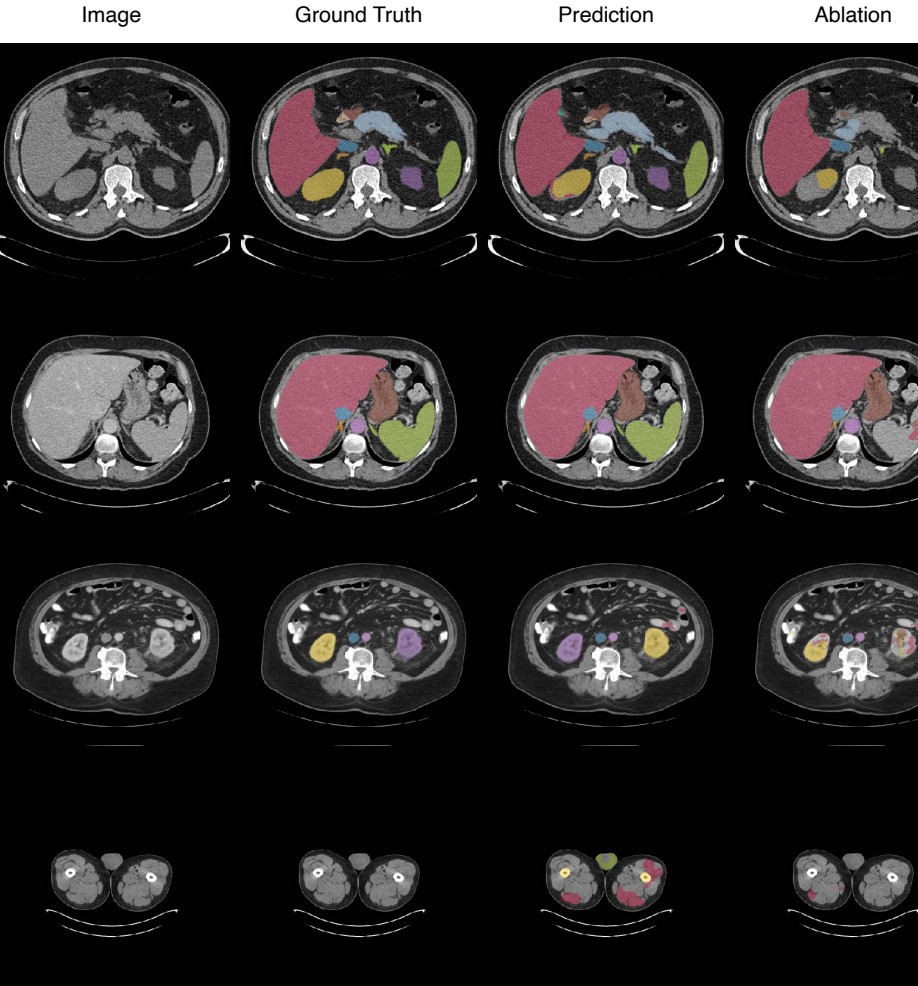

**Fig. 2.** Qualitative results of the two submitted method and the ablation on four example cases. The upper two rows show cases, where the model performs well, the lower two rows show failure cases due to left-right confusion and implausible predictions, respectively.

**Table 4.** Evaluation of segmentation efficiency in terms of the running time. Evaluation CPU platform: AMD Ryzen 9 3900X processor.

| Case ID | Image Size | Running Time (s) |
|---------|------------|------------------|
| 0005 | (512, 512, 124) | 17.6 |
| 0059 | (512, 512, 55) | 13.8 |
| 0112 | (512, 512, 299) | 22.9 |
| 0134 | (512, 512, 597) | 44.7 |
| 0135 | (512, 512, 316) | 24.0 |
| 0150 | (512, 512, 457) | 35.4 |
| 0159 | (512, 512, 152) | 22.3 |
| 0176 | (512, 512, 218) | 23.5 |

**Table 5.** Segmentation performance on the test set cohorts.

| Cohort | DSC (%) | | NSD (%) | |
|--------|---------|--------|---------|--------|
| | Avgerage | Median | Avgerage | Median |
| Asian | $73.9 \pm 12.5$ | $76.6\,(66.3, 83.6)$ | $79.3 \pm 14.1$ | $83\,(71.5, 90.4)$ |
| European | $78.2 \pm 13.5$ | $82.1\,(73.8, 87.2)$ | $82.7 \pm 14.9$ | $87\,(78.7, 92.9)$ |
| North American | $70.7 \pm 15.5$ | $75.8\,(61.8, 81.7)$ | $73.1 \pm 17.5$ | $79.2\,(62.6, 86.1)$ |

**Table 6.** Segmentation efficiency on the test set cohorts.

| Cohort | Runtime (s) | |
|--------|-------------|--------|
| | Avgerage | Median |
| Asian | $26 \pm 6.8$ | $25.5\,(20.6, 29.3)$ |
| European | $24.2 \pm 8.4$ | $24.9\,(16.9, 27.8))$ |
| North American | $20.8 \pm 9.7$ | $18.1\,(15.8, 20.7)$ |

reduce implausible predictions as described in the qualitative results section. Another limitation is the necessity for low resolution in order to comply with the inference time limit, which leads to a loss of context, especially for smaller structures.

## 5     Conclusion

In this paper, we addressed the challenge of CPU-based abdominal organ segmentation on pathological CT scans in the context of Task 2 of the FLARE 2024 challenge. Our approach to this task utilized nnU-Net, building a low-resolution configuration for efficient segmentation. We resampled the scans to an isotropic low resolution and compared this to the naive approach of using half resolution. The isotropic resolution performed significantly better compared to the half resolution. We hypothesize that the half resolution model loses most of the 3D context due to the large slice thickness of 5mm compared to 2.5mm in the isotropic case. The model complied with the time limits for inference in local testing, but might exceed these limitations during final testing on particularly large scans.

**Acknowledgements** The authors of this paper declare that the segmentation method they implemented for participation in the FLARE 2024 challenge has not used any pre-trained models or additional datasets other than those provided by the organizers. The proposed solution is fully automatic without any manual intervention. We thank all data owners for making the CT scans publicly available and CodaLab [21] for hosting the challenge platform.
The present contribution is supported by the Helmholtz Association under the joint research school "HIDSS4Health – Helmholtz Information and Data Science School for Health". Part of this work was funded by Helmholtz Imaging (HI), a platform of the Helmholtz Incubator on Information and Data Science. This work was partially supported by RACOON, funded by "NUM 2.0" (FKZ: 01KX2121) as part of the RACOON Project.

## Disclosure of Interests

The authors declare no competing interests.

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
