# OpenReview forum: "Optimizing nnU-Net with OpenVINO for Fast CPU Inference in Abdominal Organ Segmentation"
_MICCAI.org/2024/Challenge/FLARE — FLARE 2024 withMinorRevisions_

### Official Review · Reviewer_erro · 2025-01-23
**Optimizing nnU-Net with OpenVINO for Fast CPU Inference in Abdominal Organ Segmentation**

**Rating:** 8
**Confidence:** 4

**Review:**

This paper presents an efficient solution for abdominal organ segmentation in a CPU environment by optimizing the nnU-Net model using OpenVINO. The paper addresses the resource and time limitations in the FLARE 2024 challenge, proposing strategies such as low-resolution resampling, disabling test-time data augmentation, and JIT compilation optimization. These strategies significantly improve inference speed while maintaining segmentation accuracy as much as possible.
Figure 1 is too simplified and needs additional module descriptions.

---

### Official Review · Reviewer_d7JL · 2025-01-27
**Optimizing nnU-Net with OpenVINO for Fast CPU Inference in Abdominal Organ Segmentation**

**Rating:** 8
**Confidence:** 4

**Review:**

This article introduces an optimized nnU-Net method combined with the OpenVINO toolkit for rapid abdominal organ segmentation in resource-constrained CPU environments. The method achieved an average organ DSC of 76.8% and NSD of 80.5% on the online validation set, with an average inference time of 26 seconds, demonstrating its effectiveness in resource-limited clinical settings. Please add the slice number to Figure 2, e.g. Case #FLARETs_0001 (slice #123).

---

### Official Review · Reviewer_iC7m · 2025-02-18
**Optimizing nnU-Net with OpenVINO for Fast CPU Inference in Abdominal Organ Segmentation**

**Rating:** 9
**Confidence:** 4

**Review:**

This paper presents an efficient solution for abdominal organ segmentation in a CPU environment by optimizing the nnU-Net model using OpenVINO. The paper addresses the resource and time limitations in the FLARE 2024 challenge, proposing strategies such as low-resolution resampling, disabling test-time data augmentation, and JIT compilation optimization. These strategies significantly improve inference speed while maintaining segmentation accuracy as much as possible. The authors achieve an average DSC of 76.8% and NSD of 80.5% with a runtime of ~26 seconds per case on a CPU, demonstrating practical value for clinical deployment. Here are  some suggestions.

1. Resolution-Performance Trade-off: While isotropic resampling improves speed, smaller organs (e.g., adrenal glands, gallbladder) show poor DSC/NSD scores (e.g., 56.1% DSC for gallbladder), suggesting that resolution adaptation strategies (e.g., multi-scale inputs) could mitigate accuracy loss.

---

> ### Author Response · Authors · 2025-03-29
>
> I agree, that such adaptation strategies could help, but this was out-of-scope for this challenge paper.

---

### Author Response · Authors · 2025-03-29

We adapted the paper according to the reviewer comments, especially regarding figures 1 and 2.

---

### Decision · Program_Chairs · 2025-03-20

**Decision:**

Accept

**Comment:**

Please carefully address the reviewers' comments in the revision.